# DAFTED: Decoupled Asymmetric Fusion of Tabular and Echocardiographic Data for Cardiac Hypertension Diagnosis

**Jérémie Stym-Popper**[1]                                 STYMPOPPER@ISIR.UPMC.FR
**Nathan Painchaud**[2]                               NATHAN.PAINCHAUD@INSA-LYON.FR
**Clément Rambour**[1]                              CLEMENT.RAMBOUR@ISIR.UPMC.FR
**Pierre-Yves Courand**[2]                          PIERRE-YVES.COURAND@CHU-LYON.FR
**Nicolas Thome**[1]                                   NICOLAS.THOME@ISIR.UPMC.FR
**Olivier Bernard**[2]                                OLIVIER.BERNARD@INSA-LYON.FR

[1] *Sorbonne Université, CNRS, ISIR, MLIA, F-75005 Paris, France*

[2] *INSA-Lyon, Université Claude Bernard Lyon 1, CNRS, Inserm, CREATIS UMR 5220, U1294, F-69621, LYON, France*

**Editors:** Accepted for publication at MIDL 2025

## Abstract

Multimodal data fusion has emerged as a key approach in recent years for enhancing diagnosis and prognosis in many medical applications. With the advent of transformer-based methods, it is now possible to combine information from different modalities that provide complementary insights. However, most existing methods rely on symmetric fusion schemes, assuming equal importance for information carried by each modality—a strong assumption that may not always hold true. In this study, we propose an alternative fusion strategy based on an asymmetric scheme. Starting with a primary modality that offers the most critical information, we integrate secondary modality contributions by disentangling shared and modality-specific information. The proposed model was validated on a dataset of 239 patients for characterizing hypertension severity by fusing time series automatically extracted from echocardiographic image sequences and tabular data from patient records. Results show that our approach outperforms existing unimodal and multimodal approaches, achieving an AUC score over 90% - a crucial benchmark for clinical use.

**Keywords:** Multimodal fusion, transformers, tables, echocardiography, hypertension.

## 1. Introduction

Artificial Intelligence (AI) and deep learning have significantly improved computer-aided diagnosis (CADx) over the last decade (Tsehay et al., 2017; Yi et al., 2022; Xu et al., 2023). This paper focuses on the characterization of cardiac hypertension (HT). Physicians integrate complementary data from diverse sources, including time-series features derived from echocardiographic sequences and Electronic Health Records (EHRs), to build a comprehensive assessment of the patient's condition (Mancia et al., 2023). Additional measurements, such as 24-hour systolic and diastolic blood pressures (SBP/DBP), are often collected to eliminate ambiguities regarding the severity of the disease. However, this process can be burdensome for patients. In this work, we propose a method to efficiently integrate a tabular representation of minimally invasive EHR data with cardiac time series automatically extracted from apical two and four chamber views (A2C and A4C) using the segmentation framework described in (Ling et al., 2023). This approach aims to enable effective stratification of hypertension while improving patient care.

Combining heterogeneous modalities, such as tabular data and time series, is a nontrivial challenge. For tabular data, tree-based models, *e.g.,* gradient boosting (Chen and Guestrin, 2016), remain the dominant approach (Grinsztajn et al., 2022). However, a naive fusion strategy that employs XGBoost on both tabular and echocardiographic inputs results in a significant drop in performance compared to using tabular data alone, as observed in related contexts (Wang et al., 2024). This can be attributed to the asymmetry of our fusion problem: the primary modality (tabular data) serves as the main source of information, while the secondary modality (time series) provides complementary details but is insufficient for accurate diagnosis on its own. Consequently, there is a need for refined and specialized multimodal fusion methods.

In the recent wave of attentional models and transformers, significant efforts have been devoted to performing multimodal fusion for medical diagnosis. The FT-Transformer (Gorishniy et al., 2021; Zhu et al., 2023) employs a self-attention mechanism and an advanced tokenizer specifically designed for tabular data. This approach has been extended to combine multimodal tabular and echocardiographic inputs in (Painchaud et al., 2024). Recently, symmetric cross-attention has been explored in IRENE (Zhou et al., 2023), enabling each modality to be sequentially contextualized by the other, as it is classically done in vision and language models (Tan and Bansal, 2019). Although these multimodal fusion methods show improvements over the FT-Transformer trained on tabular data alone, they process the different modalities symmetrically, assuming equal relevance between them. This assumption does not align with the inherent asymmetry of our problem, where tabular data is the primary modality, and echocardiographic time series provide complementary but secondary information. Finally, an alignment loss inspired by CLIP (Radford et al., 2021) has been used in MMCL (Hager et al., 2023) to merge tabular data and medical images. While aligning tabular and echocardiographic representations is relevant in our context, applying a global alignment across all features from both modalities is overly restrictive, since tabular data contains information that is not present in echocardiographic videos.

This paper introduces a method for the characterization of cardiac hypertension that explicitly addresses multimodal fusion with asymmetric modalities, *i.e.,* , a primary source of information – tabular data – and a secondary source – time series extracted from echocardiographic image sequences. The approach shown in Fig 1 is devoted to effectively merge the primary and second multimodal inputs. Starting from a unimodal processing of each modality, we learn a relevant and structured latent space (middle in Fig 1) and introduce a fusion operator dedicated to asymmetric fusion (right in Fig 1). Our contributions can be summarized as follows:

- We separate tabular data into specific information and information shared with echocardiographic time series. This is done using specialized loss functions that differentiate information types and use label supervision for self-regularization. This method enables better multimodal data integration by organizing the latent space into shared and modality-specific attributes.

- We introduce an asymmetric fusion scheme based on interleaved cross-attention, that prioritizes one modality while gradually contextualizing it with the secondary complementary information. By emphasizing one modality and using the other for enhancement, we achieve a nuanced and effective integration of multimodal information.

## 2. Methodology: Decoupling information and fusing asymmetric modalities

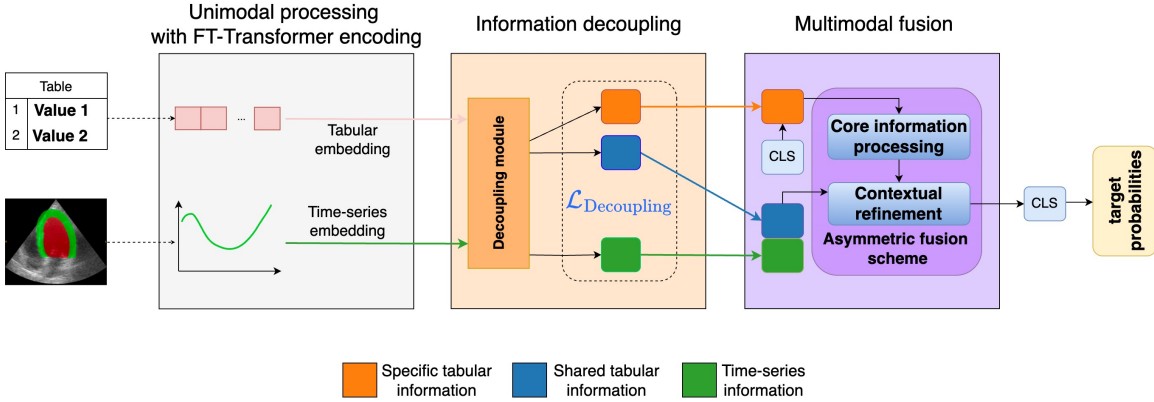

Figure 1: Overview of our model's global architecture. Tabular and echocardiographic data are first processed and tokenized separately. Each modality is encoded using a FT-Transformer (Gorishniy et al., 2021) before entering the decoupling module. The primary modality (tabular data) is then split into shared and modality-specific components. Finally, a multimodal fusion scheme is implemented, accounting for the asymmetric contribution of the two modalities. The primary modality drives core information processing, while the secondary modality provides contextual refinement.

In this section, we present the key contributions of the proposed method. First, a decoupling module re-expresses the primary modality (tabular data) into a modality-specific representation and a shared representation with the complementary modality (time series). Given the asymmetry in information content between tabular data and echocardiographic videos, where the former may include details not visually apparent in the latter, it is reasonable to consider this multimodal data as inherently imbalanced in terms of information richness. Our approach is to decouple tabular data into two components: shared features, such as left ventricular mass, which are represented in tabular data and time series, and specific tabular features, such as demographic attributes (e.g. age, sex), which are not present in time series. Our insight is to learn a space in which shared features are aligned, while keeping information from the specific and complementary information in tabular data. Secondly, the rich tabular-specific representations are re-contextualized using the shared and time-series embeddings through an asymmetric fusion scheme.

### 2.1. Decoupling module

In the multimodal framework, effectively integrating diverse data sources remains a critical challenge. A naive fusion may be suboptimal as one modality could be less informative. In our case, tabular data provide diverse and efficient information to characterize hypertension, while time series can offer additional valuable insights. To efficiently fuse these sources of information, we propose to decouple the tabular data into tabular-specific tokens and tokens that share common information with time series. To do so, an **information decoupling**

**module** (see Figure 1) is introduced after the unimodal processing of the two modalities and before the fusion scheme.

Let denote $\boldsymbol{t}$ and $\boldsymbol{s}$ the embedded tabular data and time series respectively. The dataset consists of tabular data and time series for each patient, along with their corresponding labels $y$. We further introduce $g_\theta^t$ and $g_\phi^s$, the decoupling functions that project the modalities into modality-specific and shared latent spaces. We chose to use linear projections for the decoupling functions, as this approach demonstrated superior performance compared to alternative methods we evaluated. The resulting embeddings are defined as follows: $\boldsymbol{z}_s = g_\phi^s(\boldsymbol{s})$ for time-series and $(\boldsymbol{z}_t^{sp}, \boldsymbol{z}_t^{sh}) = g_\theta^t(\boldsymbol{t})$ for specific and shared tabular modality respectively. To enforce the information decoupling, we employ the following decoupling loss which enhances the alignment of shared information representations while simultaneously distinguishing modality-specific elements:

$$l_i^{s,t} = -\log \left( \frac{\exp\{\text{sim}(\boldsymbol{z}_{s_i}, \boldsymbol{z}_{t_i}^{sh})/\tau\}}{\sum_{k=1}^N \exp\{\text{sim}(\boldsymbol{z}_{s_i}, \boldsymbol{z}_{t_k}^{sp})/\tau\}} \right), \tag{1}$$

where $\text{sim}(u,v) = u^\top v / \|u\| \|v\|$ is the cosine similarity. As illustrated in 2-a, this contrastive loss uses the projected time-series representation vector $\boldsymbol{z}_s$ as the anchor point, toward which the representation $\boldsymbol{z}_t^{sh}$ is drawn, as they encapsulate similar information. Conversely, $\boldsymbol{z}_t^{sp}$ should be mapped far from the anchor point to preserve specific information from the primary modality. To maximize this contrastive effect, we aim to minimize the **SHared-Specific Decoupling** (SHSD) loss, defined as the sum of 1 and its symmetrical counterpart:

$$\mathcal{L}_{\text{SHSD}}(\boldsymbol{z}_s, \boldsymbol{z}_t^{sh}, \boldsymbol{z}_t^{sp}) = \frac{1}{2N} \sum_{i=1}^N \left( l_i^{s,t} + l_i^{t,s} \right). \tag{2}$$

Following the approach of Yeh et al. (2022), our method removes positive pairs from the denominator of the decoupling contrastive loss, addressing the *negative-positive coupling* problem where positive samples in the denominator are inadvertently repelled.

To reinforce the overall coherency of the latent space, we introduce a secondary loss that brings closer the representations sharing the same labels, while pushing apart the embeddings of other samples in the batch:

$$r_i^{t,s} = -\frac{1}{S_i} \sum_{j=1}^N \mathbb{1}\{y_j = y_i\} \log \left( \frac{\exp\left\{ \text{sim}\left( \boldsymbol{z}_{t_i}^{sp}, \boldsymbol{z}_{s_j} \right) /\tau \right\}}{\sum_{k=1}^N \exp\left\{ \text{sim}\left( \boldsymbol{z}_{t_i}^{sp}, \boldsymbol{z}_{s_k} \right) /\tau \right\}} \right), \tag{3}$$

where $\mathbb{1}\{y_j = y_i\}$ is an indicator function and $S_i = \sum_{j=1}^N \mathbb{1}\{y_j = y_i\}$.

The expected organization of the latent space following the minimization of this loss is illustrated in 2-b. This loss acts as a regularization and ensures that specific tabular information $z_t^{sp}$ is positioned closer to the time-series representations $z_s$ of patients within the same label group.

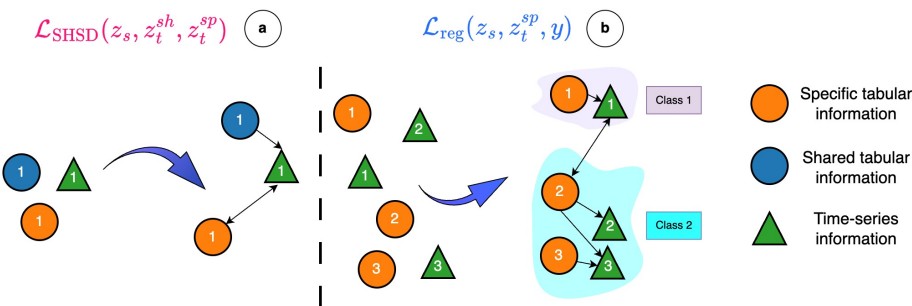

Figure 2: Decoupling module scheme in DAFTED. The SHSD loss enforces the tabular information separation by pushing the specific information away from time series while pulling the shared tabular information closer to it. Simultaneously, the regularization loss behaves similarly to the CLIP loss (Radford et al., 2021) but with label supervision (Khosla et al., 2020). It specifically pulls together the specific tabular and time-series data belonging to the same class while pushing apart information from different classes

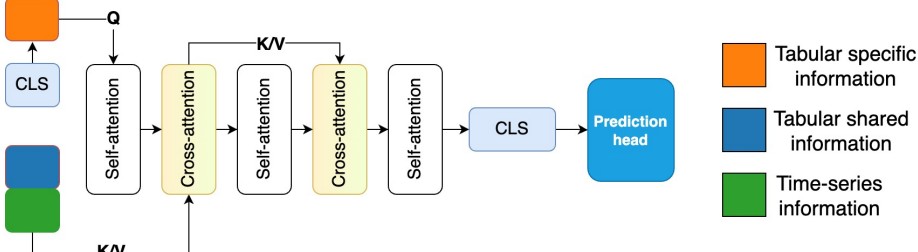

Figure 3: Asymmetric fusion scheme architecture. The tabular modality containing specific information is the prevailing modality that serves as the query tokens in all blocks, while the shared tabular and time-series information collaboratively enrich the primary modality. The cross-attention blocks employ shared weights, following Jaegle et al. (2021), strategically enforcing an asymmetric relationship between modalities. The [CLS] token serves as the class aggregate representation for the final prediction head as in Devlin et al. (2019)

Again, to equivalently update the model with respect to each modality, we use the sum of the loss 3 and its symmetrical. The regularization loss can thus be formulated as follows:

$$\mathcal{L}_{\text{reg}}(\boldsymbol{z}_s, \boldsymbol{z}_t^{sp}, y) = \frac{1}{2N} \sum_{i=1}^{N} \left( r_i^{t,s} + r_i^{s,t} \right). \tag{4}$$

Finally, the total decoupling loss denoted $\mathcal{L}_{\text{Decoupling}}$ is defined as the sum of the SHSD and regularization loss:

$$\mathcal{L}_{\text{Decoupling}}(\boldsymbol{z}_s, \boldsymbol{z}_t^{sp}, \boldsymbol{z}_t^{sh}, y) = \mathcal{L}_{\text{SHSD}}(\boldsymbol{z}_s, \boldsymbol{z}_t^{sh}, \boldsymbol{z}_t^{sp}) + \mathcal{L}_{\text{reg}}(\boldsymbol{z}_s, \boldsymbol{z}_t^{sp}, y). \tag{5}$$

To improve the merging process, we propose in the following section an asymmetric fusion scheme that leverages the decoupling module to hierarchically integrate specific tabular information as the primary modality and shared tabular information with time series as sources of contextual refinement

## 2.2. Fusion scheme: interleaved attention modules

Building on our earlier insight that modalities contribute asymmetrically to information content, we design a fusion scheme that prioritizes tabular data while using time series for contextual refinement. The Transformer architecture (Vaswani et al., 2017), leveraging self- and cross-attention mechanisms, provides an efficient framework for processing asymmetrical information by treating the most informative modality as the query $Q$, and the other as the context key $K$ and value $V$, as shown in Fig. 3.

In the fusion scheme, self-attention processes the primary tabular data, while cross-attention integrates context from shared tabular and time-series tokens. Self-attention and cross-attention blocks alternate, where the cross-attention blocks use shared weights, establishing an asymmetry between the primary tabular data and the contextual refinement from time-series and shared tabular information. A class token (Devlin et al., 2019) is appended to the specific tabular tokens at the first stage of the architecture and serves as input to a prediction head at the final stage to perform classification. The entire pipeline is optimized by minimizing the final loss defined in Eq. 6:

$$\mathcal{L}(\hat{y}, y, \boldsymbol{z}_s, \boldsymbol{z}_t^{sh}, \boldsymbol{z}_t^{sp}) = \mathcal{L}_{\text{CrossEntropy}}(\hat{y}, y) + \lambda \mathcal{L}_{\text{Decoupling}}(\boldsymbol{z}_s, \boldsymbol{z}_t^{sp}, \boldsymbol{z}_t^{sh}, y), \qquad (6)$$

where $\lambda$ is a unique hyperparameter that balances the cross-entropy and decoupling terms.

## 3. Experimental setup

**Dataset** CARDINAL is a valuable dataset combining echocardiographic image sequences from A2C and A4C views with comprehensive tabular data including demographics, lab results, and clinical exam measurements. (Ling et al., 2023). This multimodal data was collected on 239 patients at the Hospices Civils de Lyon, France, with the approval of the local ethics committee. The tabular data corresponds to 64 numerical and categorical descriptors, extracted from the EHR server. We used the hyper-tension severity (HT-severity) descriptor as the target to predict for each patient. It consists of three labels: *wht* (White Coat Hypertension), for subjects with no positive diagnosis of hypertension; *controlled*, for patients where the hypertension is managed to meet recommended blood pressure levels; and *uncontrolled*, for patients who remain above these levels. Additional information regarding the data used in our study is provided in Appendix F.

**Implementation Details** Training was conducted for 1000 epochs for DAFTED and all baseline models, the model exhibiting the lowest validation loss being selected for evaluation on the test dataset. All results utilized a temperature of $\tau = 0.1$ for the decoupling losses. We employed the Adam optimizer (Kingma and Ba, 2015) with a batch size of 128. The models were trained on 171 samples, evaluated on 20 samples, and tested on 48 samples, with a balanced distribution of the three target labels across these subsets. For the tabular modality, we selected 13 statistically independent features that are highly correlated with HT severity and easily measurable and accessible (unlike 24-hour measurements), while for the echocardiographic data, we retained the 7 measurements per view proposed by Painchaud et al. (2024), resulting in 14 times-series automatically extracted from the two apical views using the segmentation framework described in (Ling et al., 2023). The FT-Transformer was configured to follow the XTab model by Zhu et al. (2023), optimized for

tabular data processing and featuring 3 transformer blocks with 8 heads of Self-Attention (MSA) and an embedding size of 192. To ensure fair comparisons, we maintained a uniform embedding size across all models. Given the class imbalance, we used ROC AUC metrics (one-vs-rest, averaged across the three target labels) for comparison.

## 4. Results

**Baselines**  The performance of our model was compared with XGBoost, a leading algorithm for tabular data analysis, in both single-modality and multimodal scenarios. For unimodal cases, we also present results of the FT-Transformer (Gorishniy et al., 2021) trained separately on tabular and time-series data. Among unimodal models, FT-Transformer was trained separately on tabular (Tab) and time-series (TS) data. XGBoost is characterized by its complexity rather than a specific number of parameters, making its size irrelevant for comparison with other models. We evaluate several state-of-the-art models that integrate tabular and imaging clinical data within a multimodal fusion framework, namely Hager et al. (2023), Zhou et al. (2023), and Painchaud et al. (2024), alongside a naive fusion baseline that concatenates the modalities before feeding them into a two-layer perceptron.

Table 1:  Comparison of our method with SOTA models.  Mean and std are computed across 10 training runs with different seeds. FT-T refers to the FT-Transformer[3].

| Model | ROC AUC | # parameters |
|---|---|---|
| *Unimodal models* | | |
| XGBoost [1] | 87.4 | N/A |
| FT-T Tab | $85.8 \pm 4.8$ | 863K |
| FT-T TS | $52.2 \pm 2.3$ | 1.0M |
| *Multimodal fusion models* | | |
| XGBoost | 79.7 | N/A |
| MLP | $81.8 \pm 1.3$ | 391K |
| MMCL [5] | $77.4 \pm 2.1$ | 1.6M |
| IRENE [23] | $86.7 \pm 2.8$ | 102.9M |
| FT-T [11] | $88.9 \pm 1.4$ | 1.1M |
| **DAFTED (ours)** | $\mathbf{91.0 \pm 0.7}$ | 3.0M |

Table 2:  Impact of the decoupling module (Decoupling) and the asymmetric fusion scheme (Asym. fusion)

| Decoupling | Asym. fusion | ROC AUC |
|---|---|---|
| ✗ | ✗ | $89.4 \pm 2.2$ |
| ✗ | ✓ | $90.4 \pm 1.0$ |
| ✓ | ✓ | $\mathbf{91.0 \pm 0.7}$ |

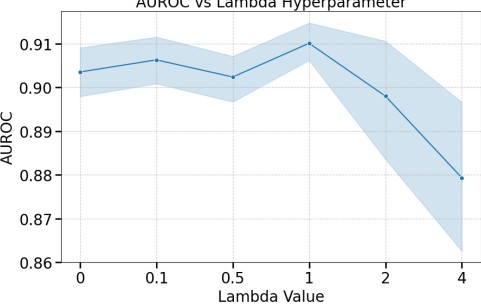

Figure 4: Sensitivity of $\lambda$ in our loss function.

**State-of-the-art comparison**  Results are presented in Table 1. XGBoost remains a strong competitor, demonstrating the best results in unimodal scenarios, while time-series data alone lacks consistent predictive power. Naive concatenation of tabular and time-series data in MLP and XGBoost models underperforms compared to unimodal tabular models or more advanced fusion approaches. Our method excels by prioritizing tabular data complemented by echocardiographic time series with a ROC AUC more than 2pt superior to

top models equally weighting both modalities. Our results were achieved without complex measurements like 24-hour systolic and diastolic blood pressure, demonstrating effective performance using simpler data acquisition methods. The paired t-tests in Appendix B show that our method significantly outperforms state-of-the-art baselines, with a 1% significance threshold for each comparison.

**Ablations and model analysis** We conducted ablation studies to evaluate the contribution of the decoupling loss and the asymmetric fusion scheme. Results presented in Table 2 show that our decoupled asymmetric fusion scheme significantly increases the performance of diagnosis by more than 2pt. We also evaluated various fusion schemes and contrastive losses within our framework. Fig. 7 compares alternative fusion schemes, while Fig. 8 explores different decoupling loss mechanisms. Classical approaches, such as InfoNCE (Sohn, 2016) and Triplet loss (Weinberger and Saul, 2009) fail to match the performance of our proposed fusion scheme, highlighting the innovative nature of our multimodal representation learning strategy. We report paired t-tests in Appendix B to quantitatively assess the significance of our method's performance. Fig. 4 investigates the influence of the hyperparameter $\lambda$ in our loss function. Results show the robustness of our method, with an optimal value of $\lambda = 1$. Fig. 5 and 6 further investigate the sensitivity of our method to the weights of the SHSD and regularization losses. This process involved setting one weight to a value of 1, while varying the other weight across a range of values from 0 to 4. Interestingly, this approach inherently includes an ablation study when one weight is set to 0 while the other remains at 1. The results demonstrate the synergistic effect of the two components of the decoupling loss. In particular, it shows the robustness of the decoupling loss across a broad range of the regularization weights, from 1 to substantially higher values. In contrast, the SHSD loss appears to be most effective in guiding the model to its optimal performance when set at a value of 1. This suggests that while the regularization component of our approach is flexible, the SHSD loss plays a crucial role in fine-tuning the model's capabilities, with its impact being most pronounced at this specific weighting. Finally, these results confirm that both components are essential for the decoupling module to be effective, with the best outcomes achieved when the weights are set to 1.

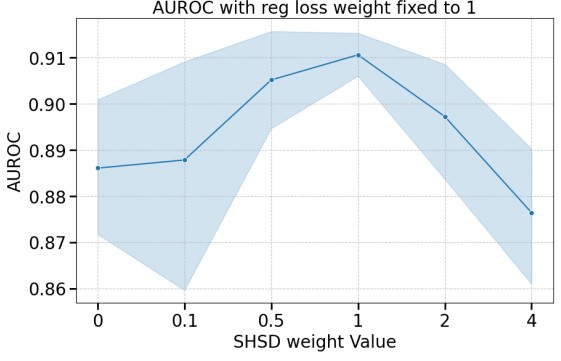

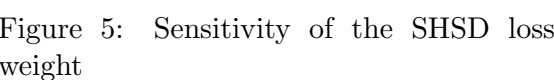

Figure 5: Sensitivity of the SHSD loss weight

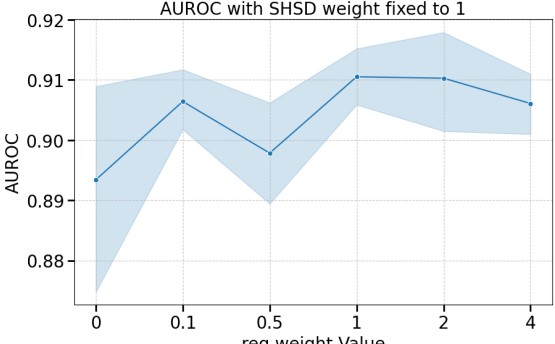

Figure 6: Sensitivity of the reg loss weight

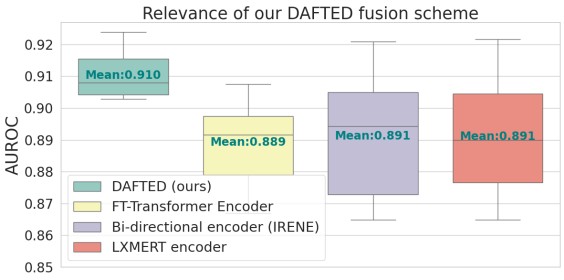

Figure 7: Impact of DAFTED fusion module vs SOTA fusion schemes

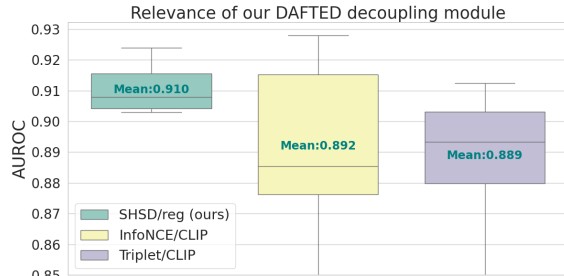

Figure 8: Impact of DAFTED decoupling module vs contrastive baselines

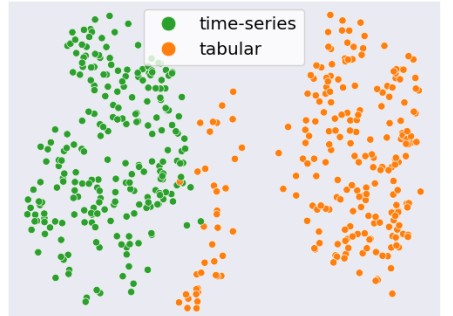

Figure 9: PacMAP representation of latent vectors before fusion without decoupling

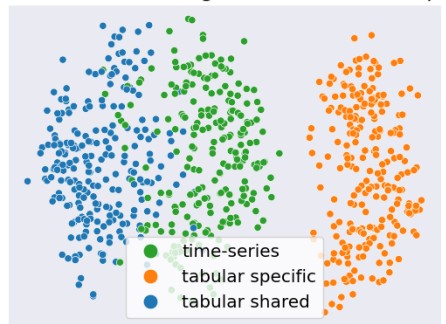

Figure 10: PacMAP representation of latent vectors before fusion with decoupling

**Latent space structuration** To visualize the structure of the latent space, we reduce the dimension of the latent vectors of each modality (tabular specific, tabular shared and time series) with the PacMAP method (Wang et al., 2021) before fusion. We observe that without decoupling the tabular information, tabular and time-series modalities are slightly intertwined (Fig. 9). Furthermore, Fig. 10 shows that the decoupling loss forces the shared tabular information to be closer to the time-series representations, while specific tabular embeddings are well separated from the time series.

## 5. Conclusion

In this study, we propose a new fusion strategy based on an asymmetric scheme. A first decoupling module separates the primary tabular modality into tokens specific to this modality and tokens shared with the secondary modality. A dedicated fusion scheme is then employed to integrate the secondary modality as a contextual refinement. Results show significant improvement compared to SOTA models, as well as stability across different parameters and architectural configurations. This study demonstrates that the characterization of HT severity can be achieved using a limited amount of easily measurable patient data, which can ultimately improve patient care.

## Acknowledgments

We would like to thank Elisa Le Maout (ARC, Hôpital Lyon Sud, Hospices Civils de Lyon, Lyon, France) for her help with data collection for the CARDINAL dataset. This research is conducted within the ORCHID project, which receives funding from the French National Research Agency (ANR) (ANR-22-CE45-0029-01). For the purpose of open access, the authors have applied a CC BY public copyright license to any Author Accepted Manuscript (AAM) version arising from this submission. We acknowledge the financial support provided by PEPR Sharp (ANR-23-PEIA-0008, ANR, FRANCE 2030).

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

## Appendix A.  Hyperparameter sensitivity test

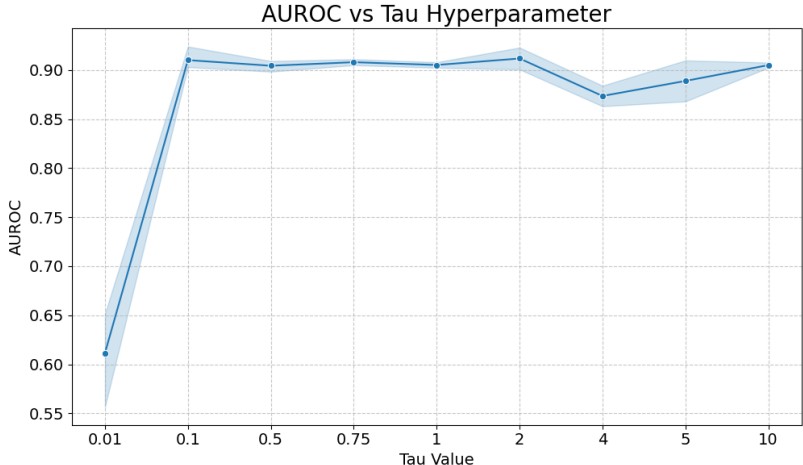

Figure 11: Hyperparameter $\tau$ sensitivity test, results averaged across 10 seeds

Figure 11 presents our analysis of the hyperparameter $\tau$, associated with the contrastive losses in equations (1) and (3). We observe that performance metrics remain relatively stable for $\tau$ values above 0.1, demonstrating our model's robustness to temperature scaling.

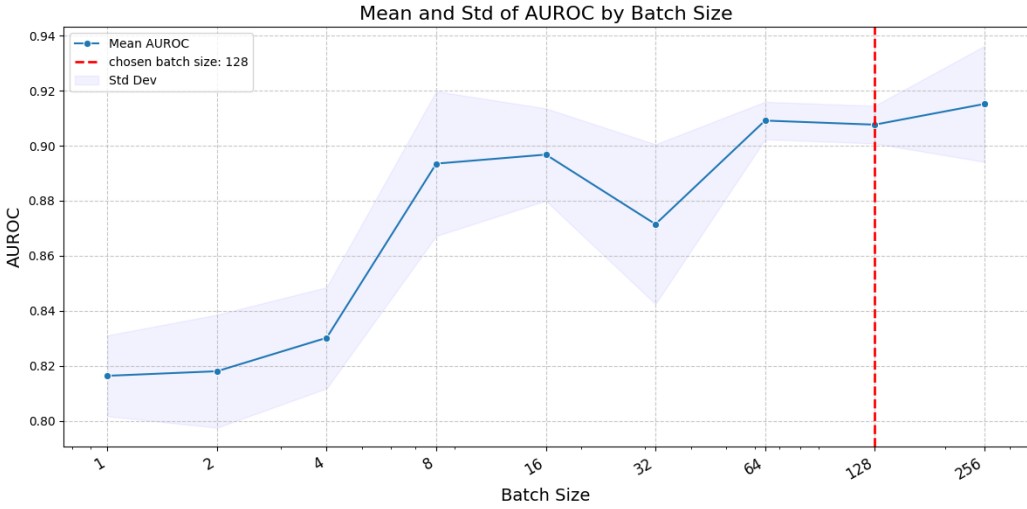

Figure 12: Batch size sensitivity test, results averaged across 10 seeds

## Appendix B.  Statistical Tests

**Paired-samples t-Test**    In Table 4, we conducted multiple paired t-tests on related samples to evaluate whether our proposed model, DAFTED, significantly outperforms the other baseline models and alternative approaches. These statistical analyses aim to assess the comparative effectiveness of DAFTED against existing methodologies in the field.

Table 3: Related-paired t-test Results Comparing DAFTED ROC metrics against baselines

| Models | Mean Difference | t-statistic | p-value |
|---|---|---|---|
| Relevance of our method against SOTA baselines | | | |
| FT-Transformer | 2.05 | 4.62 | 0.0014 |
| IRENE | 4.31 | 4.53 | 0.0014 |
| MLP | 9.22 | 23.88 | $\ll 0.001$ |
| MMCL | 13.57 | 24.01 | $\ll 0.001$ |
| Impact of our decoupling asymmetric fusion scheme (DAF) | | | |
| Our method without DAF | 2.11 | 3.46 | 0.0071 |

Table 4: Related-paired t-test Results Comparing DAFTED ROC metrics against alternative decoupling modules and fusion schemes

| Models | Mean Difference | t-statistic | p-value |
|---|---|---|---|
| Alternative fusion modules (with decoupling) | | | |
| FT-Transformer | 1.59 | 3.22 | 0.0104 |
| Bi-directional (IRENE) | 1.92 | 2.66 | 0.0259 |
| LXMERT | 1.93 | 2.87 | 0.0184 |
| Alternative decoupling modules (with our DAFTED fusion scheme) | | | |
| InfoNCE/CLIP | 1.84 | 1.93 | 0.0856 |
| Triplet/CLIP | 2.15 | 3.14 | 0.0119 |

## Appendix C. Additional results

Table 5: Other metrics to compare our model DAFTED againt baselines and SOTA models

| | AUROC | AUPRC | F1-score |
|---|---|---|---|
| XGBoost | 87.4 | 77.6 | 65.4 |
| MLP | $81.8 \pm 1.3$ | $59.8 \pm 2.0$ | $52.2 \pm 6.8$ |
| FT-Transformer | $88.9 \pm 1.1$ | $79.6 \pm 2.7$ | $68.7 \pm 3.6$ |
| **DAFTED (ours)** | $\mathbf{91.0 \pm 0.7}$ | $\mathbf{82.2 \pm 4.8}$ | $\mathbf{69.9 \pm 5.1}$ |

## Appendix D. Loss details

Table 6: Loss details

| Name | Symbol | Equation |
|------|--------|----------|
| Cross-entropy | $\mathcal{L}_{\text{CrossEntropy}}(y, \hat{y})$ | $\sum_{i=1}^{N} y_i \log(\hat{y}_i)$ |
| Shared-Specific Decoupling (SHSD) | $\mathcal{L}_{\text{SHSD}}(z_s, z_t^{sh}, z_t^{sp})$ | $l_i^{s,t} = -\log\left( \frac{\exp\{\text{sim}(z_{s_i}, z_{t_i}^{sh})/\tau\}}{\sum_{k=1}^{N} \exp\{\text{sim}(z_{s_i}, z_{t_k}^{sp})/\tau\}} \right)$ |
| Regularization (reg) | $\mathcal{L}_{\text{reg}}(z_s, z_t^{sp}, y)$ | $r_i^{t,s} = -\frac{1}{S_i} \sum_{j=1}^{N} \mathbb{1}\{y_j = y_i\} \log\left( \frac{\exp\left\{\text{sim}\left(z_{t_i}^{sp}, z_{s_j}\right)/\tau\right\}}{\sum_{k=1}^{N} \exp\left\{\text{sim}\left(z_{t_i}^{sp}, z_{s_k}\right)/\tau\right\}} \right)$ |

## Appendix E. Hyperparameters

We detail here the hyperparameter choices for training both our model and the baselines. We experimented with various values for the decoupling loss weight, temperature scale, and batch size, selecting the configuration that yielded the best performance for our model. The number of layers in the transformer unimodal encoders was kept fixed to align with state-of-the-art architectures that have demonstrated strong results across multiple tasks. For each hyperparameter, we specify the final value used in our model training.

Table 7: Hyperparameter details

| Hyperparams. | Role | Value |
|--------------|------|-------|
| $\lambda$ | Balances the decoupling loss, adjusting its weight comparing the classification cross-entropy loss | $\lambda = 1$ |
| $\tau$ | Denotes the temperature scale parameter for the contrastive decoupling losses | $\tau = 0.1$ |
| batch size | Number of samples processed per batch in the dataset | bs $= 128$ |

## Appendix F. Data specification

Table 8: List of 13 patient descriptors for the CARDINAL dataset extracted from Electronic Health Records (EHRs)

| Abbreviation | Unit/Labels | Description |
|---|---|---|
| age | years | Age |
| sbp_tte | mmHg | Systolic Blood Pressure (SBP) during TTE |
| pp_tte | mmHg | Pulse Pressure (SBP) during TTE |
| diastolic_dysfunction | 0–4 | 1 point per parameter of diastolic dysfunction: *dilated_la*, *reduced_e_prime*, *d_dysfunction_ratio*, *ph_vmax_tr* |
| pw_d | cm | Left ventricular Posterior Wall (PW) thickness at end-Diastole (D) |
| lvm_ind | g/m$^2$ | Left Ventricular Mass (LVM) indexed to BSA |
| e_e_prime_ratio | – | Ratio of E velocity over e': E/e' |
| gfr | mL/min/1.73m$^2$ | Glomerular Filtration Rate (GFR) indexed to standard body surface area |
| lateral_e_prime | cm/s | Lateral mitral annular velocity (e') |
| septal_e_prime | cm/s | Septal mitral annular velocity (e') |
| a_velocity | m/s | A-wave (active blood flow caused by atrial contraction) velocity |
| ddd | – | Defined Daily Dose (DDD) of blood pressure medication |
| la_volume | mL/m$^2$ | Left Atrial (LA) volume indexed to body surface area (BSA) |

Table 9: List of 7 patient descriptors extracted from segmentations of transthoracic echocardiogram (TTE) for the CARDINAL dataset, extracted frame-by-frame, available for apical 4 chamber (A4C) and apical 2 chamber (A2C) views

| Abbreviation | Unit/Labels | Description |
|---|---|---|
| lv_area | cm$^2$ | Surface area of the LV |
| lv_length | cm | Distance between the LV's apex and midpoint at the base |
| gls | % | Global Longitudinal Strain (GLS) |
| ls_left | % | Regional Longitudinal Strain (LS) at the base of the left wall *A4C left wall*: septum / *A2C left wall*: inferior |
| ls_right | % | Regional Longitudinal Strain (LS) at the base of the right wall *A4C right wall*: lateral / *A2C right wall*: anterior |
| myo_thickness_left | cm | Average myocardial thickness at the base of the left wall *A4C left wall*: septum / *A2C left wall*: inferior |
| myo_thickness_right | cm | Average myocardial thickness at the base of the right wall *A4C right wall*: lateral / *A2C right wall*: anterior |

