# OpenReview forum: "DAFTED: Decoupled Asymmetric Fusion of Tabular and Echocardiographic Data for Cardiac Hypertension Diagnosis"
_MIDL.io/2025/Conference — MIDL 2025 Oral_

### Official Review · Reviewer_5wTC · 2025-02-14

**Confidence:** 4
**Preliminary Rating:** 4
**Final Rating:** 5

**Summary:**

This paper proposed a multi-modality fusion model, aiming to characterize hypertension severity by combining both echocardiographic image sequences and tabular information. Overall, the proposed method was based on the CLIP loss, but also being improved by characterizing different information in tabular data, and together with label supervision. However, some improvements could be done to make it a stronger paper.

**Strengths:**

-	They borrowed the idea from the CLIP loss to apply it in a different topic, how to fuse multiple modalities with one of them being the primary modality (e.g., tabular data). They used the CLIP not only to fuse two modalities, but also separate the primary modality into two parts, specific and shared information (shared with the other modality).
-	They also add the supervision loss to make it more reasonable when dealing with a classification problem.
-	Baseline methods are fair enough.

**Weaknesses:**

-	Lack of key details of the dataset. How many positive/negative samples? How imbalance?
-	ROCAUC might not be a good metric, see below comments.
-	Some details in the loss could be discussed/improved, see below comments.
-	Ablation studies are good, but need a few more, see below comments.

**Detailed Comments:**

-	In the secondary loss with label information, did you consider using sim(z_{si}, z_{sj}) and sim(z_{ti}^{sp}, z_{tj}^{sp}) that use the same type of information? Your current loss seems to only consider cross-modality loss.
-	How do keep balance between the two losses in Equation (5)?
-	Is this experiment sensitive to the batch size?
-	In the case of class imbalance, ROCAUC might not be a good metric, because the high AUC is possibly driven by the majority. F1 score and accuracy is preferred. You may report all those metrics.
-	In Table 1 unimodal scenario, what information did XGBoost use? Tab?
-	Could you try removing different loss terms in the Ablation studies?

**Justification Of The Final Rating:**

The novelty and the clarity are both good. The proposed idea could be generalized to many other areas, especially multi-modality datasets with all different sets having both similar and complementary information.

The authors have successfully addressed my main questions, so I change my rating to Strong accept.

**Justification Of The Preliminary Rating:**

The novelty and the clarity are both good. The proposed idea could be generalized to many other areas, especially multi-modality datasets with all different sets having both similar and complementary information.

Only a few things to add to make it better.

**Questions To Address In The Rebuttal:**

I would be very curious if the terms sim(z_{si}, z_{sj}), sim(z_{ti}^{sp}, z_{tj}^{sp}), and sim(z_{ti}^{sh}, z_{tj}^{sh}) would be useful.

---

> ### Author Response · Authors · 2025-03-07
> **Answers to Questions To Address In The Rebuttal**
>
> We thank the reviewer for their interest in our paper and their insightful suggestion. We have considered the most logical approach to integrating the terms suggested by the evaluator, which would tend to group the different modalities together.
>
> In the decoupling regularization loss (equations 3 and 4, p. 4-5), the specific tabular information z_{ti}^{sp} and the time-series information z_{si} are considered as two different modalities. Therefore, our loss took inspiration from the mechanism by which CLIP aligns two modalities. In the numerator, sim(z_{si}, z_{tj}^{sp}) brings closer the two modalities when they share the same label, while the denominator normalizes the loss and pushes away the modalities when they are associated to different labels.
> Additional terms in the loss pointed out by the reviewer, such as sim(z_{si}, z_{sj}) or sim(z_{ti}^{sp}, z_{tj}^{sp}), could consolidate a hierarchy in the latent representation, favoring intra-modal closeness within hypertension severity clusters. To fulfill the reviewer request, we perform an additional experiment by including the following three additional terms into the decoupling loss:
>
> $\exp{sim(z_{si}, z_{sj})}/∑_{k} \exp{sim(z_{si}, z_{sk})}$
>
> $\exp{sim(z_{ti}^{sp},z_{tj}^{sp})}/∑_{k}\exp{sim(z_{ti}^{sp}, z_{tk}^{sp})}$
>
> $\exp{sim(z_{ti}^{sh},z_{tj}^{sh})}/∑_{k}\exp{sim(z_{ti}^{sh}, z_{tk}^{sh})}$
>
> in a manner analogous to the regularization loss. These results showed  a degradation compared to our proposed losses when adding the intra-modal terms, with a ROCAUC reaching 89.5 ± 2.1. We explain this degradation by the fact that decoupling losses offer a good balance between structuring the representation according to modality and hypertension labels, with the additional intra-modal terms destabilizing this balance.

---

> ### Author Response · Authors · 2025-03-13
> **Answers to detailed comments**
>
> **About the balance of the decoupling losses in equation (5)**: We evaluated our model with different weightings for the SHSD and regularization components of the decoupling loss from equation (5). We have added new figures (5 and 6 on p. 8) in the revised paper to illustrate the sensitivity to the loss weights hyperparameters. The results indicate that the optimal configuration assigns equal weight to both loss functions, with a factor of λ = 1. In addition, we have provided a more detailed rationale for this equal weighting in the revised version of our paper (p. 8, orange text), explaining how the two losses work synergistically and achieve optimal performance when equally weighted.
>
> **About the batch size**: To fulfill the reviewer’s request, we conducted an analysis of the impact of the batch size on performances. Results are gathered in the table below:
>
> | Batch size | AUROC |
> |------------|-------|
> | 1 | 81.6 |
> | 2 | 81.8 |
> | 4 | 83.0 |
> | 8 | 89.3 |
> | 16 | 89.7 |
> | 32 | 87.1 |
> | 64 | 90.9 |
> | 128 | 91.0 |
> | Full train set | 91.5 |
>
> This additional experiment shows that our model is robust to batch size variations, as long as batch sizes are not too small (i.e. less than 8), but appears to perform optimally for batch sizes of at least 64.
> We plan to add this result in a figure in our supplementary materials in the final version of our paper.
>
> **About additional metrics (AUPRC and F1)**: We appreciate your feedback, and to fulfill your request we conducted a new set of evaluations using AUPRC (Area Under the Precision-Recall Curve) and F1-score metrics. In the table below, we report the results of the best baselines in table 1 of our paper (p. 7) with respect to these metrics:
>
> | Model | AUPRC Avg | F1 Avg |
> |-------|-----------|--------|
> | DAFTED | 82.3 | 69.9 |
> | FT-Transformer | 79.6 | 68.7 |
> | MLP | 59.8 | 52.2 |
> | XGBoost | 77.6 | 65.4 |
>
> We observe the same trend for these metrics as the one observed for the ROC-AUC metric, with our method outperforming all baselines.
>
> **About the information used by XTab in unimodal scenario**: Indeed, in the unimodal scenario, XGBoost is trained and evaluated on the primary modality, i.e., on tabular data.
>
> **About the different loss term removals in ablation studies**: This question was addressed in the broader context of the reviewer’s question regarding the balance between the two losses. The additional figures (5 and 6 on p. 8) and text (p. 8, orange text) discussing the sensitivity to losses’ weights implicitly include an ablation study, when weights are set to 0. Performance degrades in these cases, and is optimal when both losses’ weights are set to 1. We refer the reviewer to the added text in the revision of the paper (p. 8, in orange) for our intuition on how both losses work synergistically.

---

> ### Comment · Reviewer_5wTC · 2025-03-14
>
> I sincerely thank the authors for their response and editing of the manuscript. They have addressed my main questions. Given the good quality and clarity, I would change my rating to <5: Strong accept>.

---

### Official Review · Reviewer_xAMx · 2025-02-16

**Confidence:** 4
**Preliminary Rating:** 5
**Recommendation:** Oral

**Summary:**

This paper presents a novel asymmetric multimodal fusion strategy for medical diagnosis, addressing the limitations of symmetric fusion methods that assume equal importance across modalities. The proposed method achieves strong performance, demonstrating clinical relevance for hypertension severity characterization. While dataset size and interpretability analysis could be improved, these limitations do not significantly undermine the paper's scientific contributions.

**Strengths:**

Clear motivation and problem formulation – The paper effectively highlights the limitations of symmetric fusion methods and justifies the need for an asymmetric approach.
Innovative asymmetric fusion strategy – By prioritizing a primary modality and selectively integrating secondary modality contributions, the method presents a novel fusion paradigm with strong theoretical grounding.
Strong performance – The model achieves an AUC > 90%, which is clinically significant for hypertension severity characterization.

**Weaknesses:**

Clarification in Figure 1 – The output section in Figure 1 should be explicitly labeled to ensure clarity in understanding how the predictions are generated.
Dataset limitations – While the medical dataset used is valuable, it is relatively small (239 patients) and may not fully demonstrate the generalizability of the approach. Future work could explore validation on larger or more diverse datasets.
Lack of interpretability analysis – Figure 7 primarily presents PacMAP visualizations, but additional attention maps or interpretability techniques could be provided to better illustrate the distinct contributions of the three feature types.

**Detailed Comments:**

The output section in Figure 1 is currently ambiguous. It would be helpful to explicitly label the outputs.

The paper validates its approach on a relatively small dataset (239 patients). While obtaining medical datasets is challenging, further discussion on generalizability and potential domain adaptation techniques would strengthen the paper.

Figure 7 presents PacMAP visualizations for feature representation, but the paper does not provide other interpretability methods, such as attention maps or feature importance analysis. Could the authors include attention-based visualizations to better illustrate how modality-specific and shared information contribute to the final prediction?

While the asymmetric fusion strategy is well-motivated, additional quantitative ablation studies could strengthen the claim. For instance, what happens if the secondary modality is given equal weight or if the feature disentanglement is removed?

**Justification Of The Preliminary Rating:**

This paper presents a well-motivated and innovative asymmetric fusion framework that addresses a critical limitation in existing multimodal approaches. Given its novel fusion strategy, strong experimental validation, and practical relevance, I recommend acceptance with minor revisions.

**Questions To Address In The Rebuttal:**

How does the method generalize beyond hypertension severity classification?

What is the effect of different weighting strategies for the primary and secondary modalities?

What's about the computational Efficiency?

**Special Issue:**

No

---

> ### Author Response · Authors · 2025-03-07
> **Answers to Questions To Address In The Rebuttal**
>
> We want to first thank the reviewer for the positive feedback on our work.
>
> **For the generalization beyond hypertension severity classification**: we have addressed this question in our global comment and refer the reviewer to it for further details. We agree that our method has the potential for application in other domains. In parallel to this work, we are collecting another cohort of 1,000 patients with a balanced distribution of healthy subjects and multiple heart diseases, including primary myocardial disease or coronary artery disease. This dataset will enhance the validation of our model by introducing an additional classification task and providing a larger, more diverse population.
>
> **For the weighting strategies for the primary and secondary modalities**: we are not sure what the reviewer means by 'different weight strategies' for the modalities themselves, as our model does not explicitly assign weights to the modalities. Rather, our asymmetric fusion scheme automatically learns to structure the primary and secondary modalities for the prediction task. We considered two possible interpretations of the reviewer’s comment.
>
> If the reviewer is curious about the weighting of the two decoupling loss components (equations 2 and 4), **we have added new figures (5 and 6 on p. 8) in the paper** to illustrate the sensitivity to the hyperparameter weights. These figures illustrate that our decoupling scheme exhibits significant robustness across a broad range of regularization loss weights, from 1 to substantially higher values. However, the SHSD loss appears to be most effective in guiding the model to its optimal performance when set at a value of 1. This suggests that while the regularization component of our approach is flexible, the SHSD loss plays a crucial role in fine-tuning the model's capabilities, with its impact being most pronounced at this specific weighting. Still, the best results were obtained with an equal weighting of the losses.
>
> We also considered the question might mean the reviewer is curious about the overall impact of considering one modality as primary and the other as secondary. In this case, we refer the reviewer to table 1 (p. 7), which reports results from multimodal methods that weight both modalities equally, and the **improved ablation study from table 2 (p. 7)**, which better breaks down the impact of the asymmetric fusion components.
>
> **For the computational efficiency**: we thank the reviewer for raising this question. Compared to the FT-Transformer baseline, we add a projection layer and a cross-attention module, resulting in a limited overhead (+1.9 million parameters) and bringing the total model complexity to 3 million parameters. In contrast, a state-of-the-art model such as IRENE utilizes more than 100 million parameters without achieving further improvements.
>
> Training our model over 1000 epochs takes around 30 minutes. Notably, the model consistently converges around 200 epochs, after 6 minutes of training. Additionally, the model's small size enables rapid inference times, taking around 2 minutes for the entire dataset, or less than 0.5 seconds per sample. In comparison, the state-of-the-art IRENE model requires up to 1 hour for training.

---

> > ### Comment · Reviewer_y8zr · 2025-03-10
> >
> > I thank the authors for their response. However, I respectfully disagree with the claim made in the “Computational Efficiency” section, specifically that “IRENE utilizes more than 100 million parameters without achieving further improvements.” I would like to clarify that this outcome is most likely due to the small size of the dataset used rather than an inherent limitation of the larger model itself. Typically, the motivation for scaling up a model’s size is precisely to enable improved generalization and performance when learning from larger datasets.

---

> > > ### Author Response · Authors · 2025-03-13
> > >
> > > Dear Reviewer y8zr,
> > >
> > > We agree that IRENE was designed to scale to larger datasets, and that its performance might be influenced by the size of our dataset. We have removed the sentence “without achieving further improvements” from the end of the quoted excerpt to avoid any misunderstandings. The question raised by Reviewer xAMx was focused on complexity rather than performance ( “What’s about computational efficiency?”). We aimed to highlight in our response that our approach is efficient in both training and inference, introducing only minimal overhead compared to the FT-Transformer baseline while requiring orders of magnitude fewer parameters than IRENE.

---

> > ### Author Response · Authors · 2025-03-13
> > **Answer to detailed comments**
> >
> > **About the Figure 1**: To fulfill reviewer xAMx‘s request, we will specify in Figure 1 of the final version of our paper the predicted probability vector of the three classes, i.e. “white coat syndrome”, controlled, uncontrolled.
> >
> > **About the dataset**: Please refer to the global response regarding this point.
> >
> > **About visualization and interpretability**: We agree with the reviewer that visualizing attention maps could be informative about each modality’s contribution to the final prediction. We are already considering studying these properties in our model. However, we are also aware that each design decision in these methods is important to ensure relevant visualizations. For example, different choices in how to combine attention weights across multiple attention heads or depth-wise blocks can significantly impact the final visualizations. Therefore, we consider this question fully deserving of its own exhaustive study, which we plan to address in future works.

---

### Official Review · Reviewer_y8zr · 2025-02-22

**Confidence:** 4
**Preliminary Rating:** 5
**Final Rating:** 5

**Summary:**

This paper introduces DAFTED, a novel multimodal fusion framework designed for cardiac hypertension diagnosis by asymmetrically integrating tabular data (primary modality) and echocardiographic time series (secondary modality). The core innovation lies in a decoupling module that separates tabular data into shared and modality-specific representations, followed by an asymmetric fusion scheme using interleaved cross-attention to prioritize tabular data while refining it with time-series data. Authors propose a fusion-specific contrastive lose called SHSD, outperforming other constrative approaches.

DAFTED achieves a state-of-the-art ROC AUC of 91.0% on the CARDINAL dataset consisting of 239 patients, outperforming unimodal baselines (e.g. XGBoost, FT-Transformer on tabular data) and SOTA symmetric fusion baselines (e.g., IRENE, MMCL). Authors also provide extensive ablation studies rationalizing different aspect of their methodology

**Strengths:**

This paper proposes a novel strategy for decoupling the tabular data into a modality-specific and shared (with time-series data) latent. It is followed by an innovative contrastive loss function that, in practice, outperforms the previous known loss functions.

The quality and variety of conducted experiments are satisfactory. The authors conduct rigorous experiments justifying the need for different loss functions along with statistical analysis. Disentanglement of latent embeddings before and after the fusion is a strong analysis.

The use of a transformer-based FT-Transformer backbone along with a streamlined fusion strategy results in a model with competitive performance and a manageable number of parameters.

The paper is well-structured in methodology explanations.

**Weaknesses:**

There are a couple of minor drawbacks with this paper:

Small Dataset: 239 patients limit statistical power and generalizability. External validation on larger cohorts is appreciated. I am not fully aware of this specific domain; however, if there is any other similar public dataset, it would be a good idea to include the results.

The paper introduces several loss components (e.g., SHSD loss, regularization loss) whose interactions could be more clearly elucidated. A simplified summary or schematic of how these losses jointly contribute to the latent space organization might improve clarity. Also the


There are some inconsistencies in the writing and typos and grammar issues.

**Detailed Comments:**

Provide a concise summary table that outlines each loss component (e.g., SHSD loss, regularization loss) along with its purpose and corresponding equation.

Include a brief summary or table of the key hyperparameters used (e.g., λ, temperature τ, batch size, number of epochs) and describe the tuning process.

Clarify the methodology behind the paired t-tests presented in the appendices, including assumptions and sample sizes, to support reproducibility.

What is S_{i} in equation 3? Please add the definition in the main text.

Typos and Grammer:
Page 2: "protype" → "prototype."
Page 3: Missing article in "a two-layer perceptron."

**Justification Of The Final Rating:**

As in my initial review, I find this paper presents a very interesting and novel approach to the problem. The authors have done an excellent job addressing all the reviewers’ concerns clearly and thoroughly during the rebuttal period. Therefore, I maintain my rating of “5: Strong Accept” and support nominating this work for an ORAL presentation.

**Justification Of The Preliminary Rating:**

The paper is rated as “strong accept”.  It introduces a novel approach to address the challenge of asymmetric multimodal fusion for cardiac hypertension diagnosis. The proposed method, DAFTED, is supported by extensive empirical evaluation, including ablation studies and paired statistical tests, which demonstrate statistically significant improvements over current state-of-the-art approaches. The comprehensive experimental analysis and clear presentation of the decoupling and fusion schemes provide strong evidence for the method’s effectiveness. However, the relatively small dataset size and the need for further theoretical insight into the loss components and hyperparameter selection prevent a higher rating. Addressing these concerns could further strengthen the contribution and generalizability of the work. Keeping the current score is subject to addressing some of the concerns and questions asked in the review.

**Questions To Address In The Rebuttal:**

There are not many problems with this paper, but to raise a few questions:

Can the authors provide further theoretical insight or intuition into why separating tabular data into modality-specific and shared components is particularly beneficial for this application? An explanation on how this decoupling aligns with the intrinsic asymmetry of the data could strengthen the conceptual foundation of the method.

Given that the CARDINAL dataset consists of only 239 patients, can the authors comment on the potential limitations of generalizing their findings? Have any preliminary experiments been conducted on larger or external datasets, or do they plan to validate the approach further?

The proposed decoupling loss comprises multiple components (SHSD loss and the regularization loss). Could the authors provide a clearer explanation of how these losses interact and contribute to organizing the latent space? A more detailed ablation or visualization of these interactions might clarify their individual contributions.

---

> ### Author Response · Authors · 2025-03-07
> **Answers to Questions To Address In The Rebuttal**
>
> We thank the reviewer for pointing out these questions.
>
> **For further theoretical insight**: given the asymmetry in information content between tabular data and echocardiographic videos—where the former may include details not visually apparent in the latter—it is reasonable to consider this multimodal data as inherently imbalanced in terms of information richness. Our approach is to decouple tabular data into two components: shared features, such as  left ventricular mass, which are represented in tabular data and time series, and specific tabular features, such as demographic attributes (e.g. age, sex), which are not present in times series. Our insight is to learn a space in which shared features are aligned, while keeping information from the specific and complementary information in tabular data.
>
> As stated in the introduction in submission, the decoupling goal is to go beyond a mere alignment between modalities: “applying a global alignment across all features from both modalities is overly restrictive, since tabular data contains information that is not present in echocardiographic videos”. Thus, the decoupling module separates the main tabular modality into a “shared” and a “specific” component, helping the subsequent interleaved attention module to integrate the data effectively. The distinctiveness of the tabular specific information, enforced by the decoupling module, is leveraged by the asymmetric fusion scheme, where the interleaved attention modules treat this modality as the core information. Cross-attention modules integrate both shared tabular and time-series information as contextual refinement, further leveraging the decoupling enforced by the losses.
>
> **To improve this explanation in the paper itself, we have added a paragraph at the beginning of the “Method” section, describing more clearly the link between the decoupling and the asymmetry of the data.**
>
> **For questions about the CARDINAL dataset**: We have addressed this question in our global comment and refer the reviewer to it for further details. Because of the special challenges with collecting data from echocardiography exams, we have not yet conducted preliminary experiments on larger or external datasets. However, we are most of the way done collecting a second dataset, which will include more patients (1,000) and cover an equal distribution of more cardiac diseases, such as primary myocardial disease or coronary artery disease, as well as healthy subjects. With such a dataset, we will be able to validate the generalization of our method to a larger population, as well as to more complex classification problems.
>
> **For the decoupling loss**: The decoupling loss comprises two components designed to work in tandem and is not expected to enhance performance when applied independently. The purpose of the SHSD loss (equation 1, p.4) is to separate tabular-specific information from tabular-shared information with respect to time-series information. However, on its own, the SHSD loss is insufficient to induce a useful representation of tabular-specific information, as its primary function is to push this information away from the time-series domain. To address this limitation and prevent the collapse of tabular-specific information, a regularization loss (equation 3, p.4) was introduced. It structures tabular-specific information according to classification labels, bringing samples belonging to the same class closer together and separating them from those of other classes. **The revised version of the paper now presents additional sensitivity tests (see figures 5 and 6 on p. 8) that demonstrate the synergistic effect of both decoupling loss components, with the best results achieved when both losses are weighted equally. These results validate the theoretical intuition stated above: using either component in isolation (i.e., setting one loss weight to 0) fails to achieve the same level of performance as when both are employed, demonstrating that both losses are essential for the decoupling module to be effective.**

---

> ### Comment · Reviewer_y8zr · 2025-03-10
>
> I sincerely thank the authors for diligently responding to the raised questions.
>
> At this stage, I will maintain my current rating. However, I encourage the authors to comprehensively address all the reviewers’ comments. In particular, I support reviewer 5wTC’s suggestion regarding evaluating the models with additional metrics beyond ROC-AUC. Given that ROC-AUC can occasionally yield overly optimistic results when negative samples significantly outnumber positive ones, metrics such as Precision-Recall (PR) AUC and F1-score would further enhance the clarity and robustness of the presented results.
>
> I will revisit this discussion section before the end of the discussion period to determine my final rating. My final evaluation will partially depend on how effectively the authors address the concerns mentioned above.

---

> > ### Author Response · Authors · 2025-03-13
> >
> > Dear reviewer y8zr,
> >
> > We appreciate your interest in our paper and how you took notice of our efforts to address the reviewers’ most pressing questions. Given the tight deadline to address the reviewers' questions in the rebuttal, we initially prioritized the items listed under "Questions To Address In The Rebuttal." However, with the additional time provided during the discussion period, we have now included new responses to address the remaining comments. We have provided further clarifications where needed and conducted additional experiments to meet the reviewers' requests. Additionally, we have outlined further improvements planned for the final version of our paper.

---

> ### Author Response · Authors · 2025-03-13
> **Answers to detailed comments**
>
> **About the concise summary table that presents the loss components**:
>
> The deadline to edit the paper as part of the rebuttal has passed, but we plan to add this summary table in the supplementary materials of the camera-ready version.
> | Loss Name       | Notation                                      | Formulation | Description |
> |-----------------|-----------------------------------------------|-------------|-------------|
> | Cross-entropy   | $\mathcal{L}_\mathrm{CrossEntropy}(\hat{y}, y)$ | $\sum_{i=1}^N y_i \log(\hat{y}_i)$       | Measures the cross-entropy between predicted labels and true labels |
> | SHSD            | $\mathcal{L}_\mathrm{SHSD}(z_s, z_t^{sh}, z_t^{sp})$  | $l_i^{s,t} = -\log{\frac{\exp{\mathrm{sim}(z_{s_i}, z^{sh}{t_i})/\tau}}{\sum_{k=1}^N\exp{\mathrm{sim}(z_{s_i},z^{sp}_{t_k})/\tau}}}$ | Separates tabular-specific information from tabular-shared information with respect to time-series information |
> | Regularization  | $\mathcal{L}_\mathrm{reg}(z_s, z_t^{sp}, y)$ | $r^{t,s}i = - \frac{1}{S_i}\sum_{j=1}^N\mathbb{1}{(y_j = y_i)}\log\left( \frac{\exp{\mathrm{sim}( z^{sp}{t_i} , z{s_j} )/ \tau}}{\sum_{k=1}^N \exp{\mathrm{sim}( z^{sp}{t_i} , z{s_k} )/ \tau}} \right)$       | Structures tabular-specific information according to classification labels, bringing samples belonging to the same class closer together and separating them from those of other classes |
>
> **About the table of the key hyperparameters**: Like for the preceding comment, we will include these details in the supplementary materials of the camera-ready version.
>
> **About $S_i$ in equation (3)**: We have added the definition of $S_i=\sum_{j=1}^N\mathbb{1}{(y_j = y_i)}$ in orange just below equation 3 in the revised version of our paper (p. 4). $S_i$ represents the number of samples in the training dataset belonging to the same class as the sample i.
>
> **Typos and Grammar**: Page 2: "protype" → "prototype." Page 3: Missing article in "a two-layer perceptron."
> Answer: Comment on typos and grammar: We have taken into account the required changes and have included them in the revised version of the paper.

---

> > ### Comment · Reviewer_y8zr · 2025-03-14
> >
> > I sincerely thank the authors for thoroughly addressing all reviewer comments. I maintain my strong accept recommendation and support nominating this paper for an ORAL presentation.

---

### Author Response · Authors · 2025-03-07

We thank the reviewers for their constructive feedback and positive comments. Several reviewers highlighted the use of a relatively small dataset in our study, raising concerns about the generalizability of our model.

Our study relies on an internal dataset named CARDINAL, which includes both echocardiographic image sequences and patient data extracted from electronic health records. Compared to other imaging modalities (e.g. MRI) where data is centralized and structured on PACS, ultrasound exams are performed in point-of-care settings and their records are not nearly as structured. Thus, collecting imaging exams requires lots of manual intervention, which makes it seriously more difficult to scale to large, multi-center datasets.
Additionally, data routinely collected to guide treatment for hypertension, especially images, is typically not of good enough quality to perform the kind of exhaustive retrospective analysis we perform. Many potentially relevant variables may not be systematically recorded, and even when they are, missing data is common due to acquisition issues (e.g., patient non-cooperation). Given these challenges, the CARDINAL dataset is unique in both its scope and the completeness of the data collected. Since it was built from a research cohort, data acquisition followed a more rigorous and comprehensive protocol than in routine clinical practice, resulting in a richer dataset with fewer missing values. To our knowledge, no comparable datasets exist, let alone larger or multi-center ones.

This work serves as a pilot study, providing an initial validation of an innovative asymmetrical fusion scheme with information decoupling. However, we recognize the need for further validation on a larger dataset and across different classification tasks. To address this, we are currently developing a second dataset, ORCHID, which will include 1,000 patients and is designed to facilitate the automatic classification of several challenging-to-diagnose heart diseases by integrating both echocardiographic images and patient data. This dataset is currently under construction, and we plan to use it in the coming months to assess the generalization capability of our approach.

All modifications to the manuscript have been documented in the rebuttal. Changes in the manuscript are highlighted in orange.

---

### Author Rebuttal · Authors · 2025-03-07

**Rebuttal:**

New revised version of our paper:

Multimodal data fusion has emerged as a key approach in recent years for enhancing diagnosis and prognosis in many medical applications. With the advent of transformer-based methods, it is now possible to combine information from different modalities that provide complementary insights. However, most existing methods rely on symmetric fusion schemes, assuming equal importance for information carried by each modality—a strong assumption that may not always hold true. In this study, we propose an alternative fusion strategy based on an asymmetric scheme. Starting with a primary modality that offers the most critical information, we integrate secondary modality contributions by disentangling shared and modality-specific information. The proposed model was validated on a dataset of 239 patients for characterizing hypertension severity by fusing time series automatically extracted from echocardiographic image sequences and tabular data from patient records. Results show that our approach outperforms existing unimodal and multimodal approaches, achieving an AUC score over 90% - a crucial benchmark for clinical use.

**Supporting Material:**

/attachment/29f217a1e1b884ed2aaa3abc156d77c9c876263e.pdf

---

### Meta-Review · Area_Chair_nMAC · 2025-03-20

**Recommendation:** Accept (Oral)
**Confidence:** 5

**Metareview:**

All reviewers have rated this paper favourably and the authors and reviewers have engaged actively during the discussion to address any remaining comments. Upon reading the work, it comes across as an innovative and compelling approach to address multimodal fusion, particularly with tabular data involved, which can often be quite heterogenous.

The ranking of the paper is also quite high with three strong accepts, due to which I would like to recommend it for publication as well as an oral presentation.